# Role of Microbiota-Derived Metabolites in Alcoholic and Non-Alcoholic Fatty Liver Diseases

**DOI:** 10.3390/ijms23010426

**Published:** 2021-12-31

**Authors:** Ji-Won Park, Sung-Eun Kim, Na Young Lee, Jung-Hee Kim, Jang-Han Jung, Myoung-Kuk Jang, Sang-Hoon Park, Myung-Seok Lee, Dong-Joon Kim, Hyoung-Su Kim, Ki Tae Suk

**Affiliations:** 1Department of Internal Medicine, Hallym University Sacred Heart Hospital of Hallym University Medical Center, 22, Gwanpyeong-ro 170 beon-gil, Dongan-gu, Anyang-si 14068, Korea; miunorijw@hallym.or.kr (J.-W.P.); sekim@hallym.or.kr (S.-E.K.); 2Institute for Liver and Digestive Diseases, Hallym University, Chuncheon-si 24252, Korea; nylee95@snu.ac.kr (N.Y.L.); jungheekim@hallym.or.kr (J.-H.K.); con2000@hallym.or.kr (J.-H.J.); mkjang@kdh.or.kr (M.-K.J.); sanghoon@hallym.or.kr (S.-H.P.); leemsmd@hallym.or.kr (M.-S.L.); djkim@hallym.or.kr (D.-J.K.); 3Department of Internal Medicine, Chuncheon Sacred Heart Hospital of Hallym University Medical Center, 77, Sakju-ro, Chuncheon-si 24253, Korea; 4Department of Internal Medicine, Dongtan Sacred Heart Hospital of Hallym University Medical Center, 7, Keunjaebong-gil, Hwaseong-si 445-907, Korea; 5Department of Internal Medicine, Kangdong Sacred Heart Hospital of Hallym University Medical Center, 18, Cheonho-daero 173-gil, Gangdong-gu, Seoul 05355, Korea; 6Department of Internal Medicine, Kangnam Sacred Heart Hospital of Hallym University Medical Center, 1, Singil-ro, Yeongdeungpo-gu, Seoul 07441, Korea

**Keywords:** alcoholic liver disease, non-alcoholic fatty liver disease, gut microbiota, gut–liver axis, dysbiosis, metabolites

## Abstract

Chronic liver disease encompasses diseases that have various causes, such as alcoholic liver disease (ALD) and non-alcoholic fatty liver disease (NAFLD). Gut microbiota dysregulation plays a key role in the pathogenesis of ALD and NAFLD through the gut–liver axis. The gut microbiota consists of various microorganisms that play a role in maintaining the homeostasis of the host and release a wide number of metabolites, including short-chain fatty acids (SCFAs), peptides, and hormones, continually shaping the host’s immunity and metabolism. The integrity of the intestinal mucosal and vascular barriers is crucial to protect liver cells from exposure to harmful metabolites and pathogen-associated molecular pattern molecules. Dysbiosis and increased intestinal permeability may allow the liver to be exposed to abundant harmful metabolites that promote liver inflammation and fibrosis. In this review, we introduce the metabolites and components derived from the gut microbiota and discuss their pathologic effect in the liver alongside recent advances in molecular-based therapeutics and novel mechanistic findings associated with the gut–liver axis in ALD and NAFLD.

## 1. Introduction

Globally, chronic liver disease, which is one of the most common medical conditions, affects approximately 840 million people and accounts for 2 million deaths per year [1]. Chronic liver disease is characterized by the progressive deterioration of liver functions, including the production of clotting factors and other proteins, the detoxification of harmful products of metabolism, and the excretion of bile. In chronic liver disease, a continuous process of inflammation, destruction, and regeneration of liver parenchyma leads to fibrosis and cirrhosis. The initial conditions or factors causing chronic liver disease include viral hepatitis, fatty liver (alcoholic or non-alcoholic), autoimmune diseases, and genetic and metabolic disorders. The continuation of the initial insult ultimately leads to decompensated liver cirrhosis; however, the cessation of the insulting factor has still been reported to result in the progression of liver disease [2]. To determine what influences this progression in addition to the primary cause of chronic liver disease, investigators have evaluated the changes in gut microbial composition, the relationship between these changes and the different causes of liver disease, and the relevance of these changes to disease progression.

Trillions of microorganisms (bacteria, protozoa, archaea, fungi, and viruses) live in the human gastrointestinal tract [3]. This microbiota has been known to play beneficial roles in the body, such as immunomodulation to prevent pathogen colonization and nutrient digestion and absorption; therefore, conditions linked to changes in the microbiota have a fundamental impact on host physiological and pathological processes [4,5]. The regulating potentiality of the gut microbiota is not confined to the intestine; it is affiliated with several distant organs, such as the kidneys, brain, cardiovascular system, bone system, and liver [6]. The gut–liver axis refers to the close bidirectional connection between the intestine and the liver via the portal vein, biliary tract, and systemic circulation.

The gut microbiota can generate bioactive metabolites from endogenous (bile acids) and exogenous (diet and environmental) substrates, and the metabolites can be transported to the liver through the venous branches of the portal vein [7]. These microbial metabolites not only interact with host signal transduction pathways in the intestine but also reach the liver through the portal vein. This review article focuses on the most recent advances in our understanding of dysbiosis-related metabolites and treatments for alcoholic liver disease, non-alcoholic fatty liver disease, and liver cirrhosis.

## 2. Gut Microbiota and Dysbiosis

The composition and number of bacteria vary according to their location in the gastrointestinal tract. The stomach and duodenum harbor 10–10^3^ bacteria per gram of intestinal content, and 10^4^–10^7^ and 10^11^–10^12^ bacterial numbers are found in the small and the large intestines, respectively. The highest bacterial levels are found in the large intestine [8]. Nearly 90% of bacteria belong to two major phyla, *Firmicutes* (Gram positive) and *Bacteroidetes* (Gram negative), followed by two minority phyla, *Proteobacteria* and *Actinobacteria*, and the rest belong to *Fusobacteria* and *Verrucomicrobia* [9,10]. The phyla contain one or more classes that comprise orders that in turn encompass families, genera, and species of bacteria. The relationship between the two major phyla, known as the ratio of *Firmicutes* to *Bacteroidetes*, has been associated with individual susceptibility to disease states, including obesity [11]. In addition, significant pathogens, such as *Escherichia coli*, *Campylobacter jejuni*, *Salmonella enterica*, *Vibrio cholerae*, and *Bacteroides fragilis*, exist in the human colon, but normally at very low levels (<0.1% gut microbiome) [12,13]. The combination of a low abundance of pathogens and a high abundance of essential genera, such as *Bacteroides, Prevotella*, and *Ruminococcus*, indicates a healthy state for the gut microbiota [14]. A stable cellular composition consisting of the dominant phyla *Bacteroidetes*, *Firmicutes*, *Actinobacteria*, and *Proteobacteria* is very important for the proper function of the gut microbiota [15].

Shifts to an “abnormal” microbiota, such as a loss of keystone taxa, pathogen proliferation, and changes in metabolic capacity, are defined as dysbiosis [16]. Dysbiosis can be characterized by a loss of beneficial bacteria, an expansion of potentially harmful organisms, and/or a loss of overall microbial diversity [17]. A growing number of diseases, such as inflammatory bowel diseases, metabolic disorders, autoimmune diseases, and neurological disorders, are reported to be associated with intestinal dysbiosis [18,19,20,21]. The leading factors affecting the composition of the gut microbiota include diet, various drugs, the intestinal mucosa, the immune system, and the microbiota itself. Many triggering factors, such as oxidative stress, bacteriophage induction, and the secretion of bacterial toxins, are associated with shifts in the microbiota to the point of dysbiosis.

## 3. Gut Microbiome and Metabolites

The gut microbiota has a cell number similar to that of humans. Furthermore, the combined genomes of the gut microbiota—the microbiome—contains 450-fold more genes than are encoded in the human genome [22]. The gut microbiota genomes encode functions and metabolic pathways that are involved in diverse host biological processes, such as metabolism, nutrition, and immunity [23,24,25]. The microbiome is defined as the assembly of microbes and their genomic components, plus the products of the microbiota and the host environment [26,27]. Advanced technologies, including 16S rRNA amplicon sequencing, shotgun metagenomic sequencing, and other multi-omic approaches, have led to the identification of the functional characteristics beyond simply profiling microbiota compositions. These methods make it possible to directly examine the phylogenetic markers, genes, transcripts, proteins, or metabolites from the samples [28]. Especially, metatranscriptomics, metaproteomics, and metabolomics are used to characterize the functional and metabolic activities of the microbiome [29,30,31].

Although the microbiota generates numerous metabolites, the key microbial metabolites include short-chain fatty acids (SCFAs) and bile acids, as well as recently identified amino acid-derived metabolites (trimethylamine-N-oxide (TMAO), indole) and endogenous ethanol. These metabolites are either derived from bacteria metabolisms, such as tryptophan, or host molecules modified by bacteria, such as bile acids. In addition to metabolites, gut-derived microbe-associated molecular patterns (MAMPs), especially pathogen-associated molecular patterns (PAMPs), may provoke or exacerbate innate immune responses in the liver. MAMPs are essential structures for the microbes and include pathogens and nonpathogenic microorganisms. PAMPs include: microbial molecular structures, such as Gram negative-related lipopolysaccharide (LPS); Gram-positive-bacteria-related lipoteichoic acid and peptidoglycan; lipoglycans, lipoarabinomannan, lipopeptides, and lipomannans from mycobacteria; zymosan from yeast; and DNA from viruses and bacteria [32,33]. PAMPs are recognized by pattern recognition receptors (PRRs). In humans, Toll-like receptors (TLRs) constitute the main family of PRRs. TLRs are expressed by hepatic stellate cells, liver parenchymal cells, such as hepatocytes, and cholangiocytes, as well as a wide variety of immune cells, including resident and circulating macrophages, dendritic cells, and neutrophils. The cellular localizations of TLRs are different, but their activation leads to the common signal transduction pathways promoting the expression and release of several pro-inflammatory cytokines, such as Tumor necrosis factor (TNF)-α, interleukin (IL)-1β, IL-6, and interferons.

Gut barrier dysfunction results in the translocation of microbes and microbially produced metabolites and products from the gut lumen to the portal vein and systemic circulation. In the hepatic sinusoid, gut microbiota-derived metabolites and PAMPs trigger a downstream complex signaling that is related to toxicity, inflammation, and gene expression responses through the PRRs. These signaling responses can lead to metabolic alterations in the liver and eventually direct the progression of chronic liver disease.

In the next section, the dysbiosis and microbe-derived metabolites will be examined through the lens of their biological significance as they are related to alcoholic liver disease, non-alcoholic fatty liver disease, and liver cirrhosis.

## 4. Alcoholic Liver Disease

### 4.1. Dysbiosis and Microbe-Derived Metabolites in Alcoholic Liver Disease

Alcohol consumption is one of the main causes of chronic liver disease and liver-related deaths worldwide. Alcoholic liver disease clinically presents from simple steatosis to steatohepatitis, eventually progressing to fibrosis and cirrhosis. During disease progression (from steatohepatitis to precirrhosis to cirrhosis), a shift in the gut microbiota composition occurs [34]. The gut microbial signature of patients with alcohol use disorder and alcoholic liver disease can be found in Table 1.

The development of small intestinal bacterial overgrowth (SIBO) and dysbiosis in the setting of alcoholic liver disease has been reported [39]. Alcohol consumption/feeding and alcoholic cirrhosis are associated with a decrease in the *Lactobacillus* species [40,41,42,43]. The *Lactobacillus* species are considered to be “good bacteria” and help suppress pathogens within the *Enterobacteriaceae* family, such as *Salmonella* or *Shigella*, by producing bacteriocins, such as antibiotics. Their peroxidase production contributes to inhibiting other bacteria [43,44]. The *Lactobacillus* species protect the host from pathogenic and invasive bacteria by adhering to intestinal epithelial cells [45,46,47].

Their fermentation products include SCFAs. SCFAs are the most plentiful bacterial metabolites derived from intestinal bacterial fermentation of indigestible carbohydrates or dietary fibers, and SCFAs are chiefly composed of acetate, propionate, and butyrate. The vital roles of SCFAs are to supply energy and nutrition to intestinal epithelial cells [48] and to maintain the integrity of the intestinal barrier and mucosal immune tolerance. Chronic ethanol administration reduces the biosynthesis of not only SCFAs but also saturated long-chain fatty acids (SLCFAs) in intestinal bacteria, which contributes to gut barrier dysfunction. Chen et al. reported that supplementation with SLCFAs maintained intestinal eubiosis and reduced ethanol-induced liver injury in mice [49]. Saturated fatty acids do not act directly on the intestine to stabilize its barrier. Instead, *Lactobacilli* metabolize SLCFAs to promote their expansion. Saturated fatty acids were reported to serve as vitamin B substitutes and promote the growth of the *Lactobacillus* species [50].

Elevated plasma endotoxin levels are observed in humans and animals with alcoholic liver disease [51,52,53]. Endotoxins, also called LPS, are components of the outer membrane of the cell wall of Gram-negative bacteria. Increased endotoxemia correlates with increased intestinal permeability [54]. Gut barrier dysfunction permits endotoxins to cross the gut barrier and reach the liver by entering the blood stream. In the liver, endotoxins interact with and activate Kupffer cells, which produce superoxide and TNF-α, leading to liver damage [55]. Ethanol can also induce a leaky gut through the disruption of epithelial tight junctions, resulting in bacterial translocation [34]. The endotoxin or peptidoglycan passage provokes cytokine release by stimulating hepatic receptors, such as TLR, ultimately leading to hepatic fat disposition and inflammation [34]. Additionally, endogenous ethanol produced by bacteria increases intestinal permeability and eventually deteriorates microbial translocation.

In mice, chronic ethanol feeding resulted in an increased bile acid pool and lower intestinal farnesoid X receptor (FXR) signaling. The primary bile acids, chenodeoxycholic acid (CDCA) and cholic acid (CA), are synthesized in the liver through the oxidation of cholesterol. Almost all of the primary bile acids are secreted into the gut, recirculated to the liver by the portal vein and reused by the liver, while the remaining 5% are transformed by the gut microbiota into secondary bile acids, lithocholic acid (LCA) and deoxycholic acid (DCA) [56]. The secondary bile acids are more hydrophobic and thus more toxic for the intestinal and hepatic epithelial cells [57,58,59,60]. The enterohepatic circulation of bile acids is extremely important for gut eubiosis. Disturbance of the normal intestinal microbiota can affect bile acid metabolism, increase the degree of secondary bile acid conversion, and thus lessen the rate of primary bile acid reabsorption [56].

Bile acids are recognized by the FXR, expressed in hepatocytes and enterocytes. FXR contributes to the generation of antimicrobial molecules in intestinal epithelial cells and prevents intestinal barrier dysfunction [61,62]. In the intestine, an FXR activated by conjugated bile acids induces endocrine hormone fibroblast growth factor (FGF)-15/19 (FGF-15 in rodents, FGF-19 in humans). FGF-15/19 decreases the transcription of cytochrome P450 enzyme 7a1 (CYP7A1) in hepatocytes, thereby suppressing de novo bile acid synthesis [63,64]. However, gut dysbiosis induced by chronic ethanol consumption results in low FGF15 plasma levels, increased hepatic CYP7A1 expression with unbalanced bile acid homeostasis, and altered liver metabolism. The signaling pathway participating in primary bile acid reabsorption accompanies the activation of FXR, which leads to the downregulation of bile acid synthesis, increased bile acid clearance, and the production of antimicrobial peptides in the lumen [56,65]. Therefore, decreased utilization of this pathway drives the intestinal environment to be more susceptible to bacterial overgrowth.

Ethanol-associated dysbiosis reduces the levels of indole-3-acetic acid (IAA), one of the bacterial tryptophan catabolites. Tryptophan is an essential amino acid for humans and is present in a variety of foods, such as cruciferous vegetables, red meat, fish, cheese, beans, and eggs [66]. Commensal bacteria can catabolize tryptophan into indole via the action of tryptophanase. IAA is a microbiota-derived ligand of the aryl hydrocarbon receptor (AHR), which regulates the expression of IL-22. IL-22 regulates the expression of regenerating islet-derived 3 gamma (REG3G), a c-type lectin produced by intestinal epithelial and Paneth cells. REG3G defends against pathogens and maintains the spatial segregation of the microbiota and the host [67]. REG3G, mainly expressed in the small intestine, regulates the intestinal immune response against pathogens and sustains the homeostasis of commensal microbes [68]. IL-22 is a cytokine primarily produced by RORγt^+^ type 3 innate lymphoid cells (ILC3s) in the gut during homeostasis [69]. The microbiota can regulate IL-22 production through the metabolites produced from tryptophan catabolism, called indoles [70]. Indoles are known to help reinforce the integrity of the intestinal barrier and are considered to be a favorable chemical signal within microbe–host interactions [71].

### 4.2. Microbiota-Changing Interventions in Alcoholic Liver Disease

The gut microbiome can be modulated with diet, pre-, pro-, or antibiotics, and fecal microbiota transplantation (FMT). Probiotics are a group of nonpathogenic, beneficial microorganisms that function to modulate and maintain a stable intestinal environment and restore microecological balance [72]. In animal studies, probiotics, such as *L. rhamnosus* or VSL#3 (a probiotic mixture, containing *Bifidobacterium breve*, *B. longum*, *B. infantis*, *L. acidophilus*, *L. plantarum*, *L. paracasei*, *L. bulgaricus*, and *Streptococcus thermophilus*) have been shown to ameliorate alcohol-induced liver inflammation and gut leakiness [73,74]. In addition, *Pediococcus pentosaceus* alleviated ethanol-induced liver injury by reversing gut microbiota dysbiosis, regulating SCFAs metabolism in an animal study [75]. Prebiotics are nondigestible food ingredients. They act to help the gut peristalsis and stimulate the growth or activity of specific species of intestinal bacteria. Fructo-oligosaccharides are commonly regarded as a type of prebiotic substance that stimulates the growth of beneficial gut microbes. A previous study showed that the use of prebiotics improved alcohol-induced liver damage in mice by increasing the level of antimicrobial protein REG3G and reducing intestinal bacterial overgrowth [40]. Ferrere et al. reported that intestinal microbiota manipulation via FMT or prebiotic treatment (pectin, a fiber present in fruits) restored *Bacteroides* levels in mice and thus prevented liver damage by alcohol [76]. In a human study, FMT in the case of severe alcohol hepatitis improved the three-month survival rate compared to other treatment groups (corticosteroid/nutrition support only/pentoxifylline) and produced favorable gut microbial changes [77].

Recently, the effectiveness of “targeted” microbiome-directed interventions, including bioengineered commensals, drugs targeting selected microbial metabolism, and phage therapy, has been reported mainly through animal studies (Table 2).

The pharmacologic manipulation of luminal SCFAs with the prodrug tributyrin (glyceryl tributyrate) showed a protective effect against the gut injury caused by ethanol exposure in mice [78,79]. Another study evaluated the effects of IL-22-producing bioengineered bacteria on ethanol-induced liver disease in mice [82]. IL-22, a cytokine mainly expressed by ILC3s, regulates the production of REG3G lectins in the gut during homeostasis [69]. C-type lectins participate in the intestinal immune response against pathogens and retain the homeostasis of commensal microbes [68]. Hendrikx et al. found that IL-22-producing bioengineered bacteria induce the expression of REG3G to reduce ethanol-induced steatohepatitis [82]. Wang et al. also reported that a TLR7 ligand, 1Z1, ameliorated alcohol-associated liver injury via the induction of IL-22 [83]. TLR7 signaling has been shown to be protective against liver fibrosis in mice [84]. TLR7 is mainly expressed in immune cells, such as macrophages, dendritic cells, and B cells. TLR7 signaling induced IFN-α production in dendritic cells, followed by IL-1 receptor antagonist induction in Kupffer cells. An IL-1 receptor antagonist suppressed IL-1-induced hepatic stellate cell activation, resulting in the inhibition of liver fibrosis [84]. Additionally, Wang et al. found that TLR7 signaling upregulated IL-22 [83]. Another study showed that the exogenous administration of IL-22 had a profound effect on tissue repair by promoting proliferation and inhibiting apoptosis in hepatocytes of mouse models of alcoholic hepatitis [85]. Bacteriophage therapy is an intervention that targets the microbiome. Bacteriophages are viruses that can specifically infect and kill bacteria, and 10^15^ types of phage are present in the human intestine [86]. Phages are generally very specific to the bacterial subtypes and can selectively infect specific bacteria; thus, a similar effect to knocking down specific bacteria is expected. In a study of the therapeutic effects of bacteriophages in alcoholic liver disease, Duan et al. found that when cytolytic *E. faecalis* was targeted by bacteriophages, there was decreased cytolysin secretion and an amelioration of alcohol-induced liver injury in mice [80]. This result suggests that phage therapy could be a treatment option for alcoholic liver disease by precisely modulating the intestinal microbiota.

## 5. Non-Alcoholic Fatty Liver Disease

### 5.1. Dysbiosis and Microbe-Derived Metabolites in Non-Alcoholic Fatty Liver Disease

Non-alcoholic fatty liver disease (NAFLD) is primarily characterized by excessive fat accumulation in hepatocytes. Although NAFLD has many similar features to alcoholic liver disease, NAFLD develops without the consumption of toxic levels of alcohol [87]. NAFLD is one of the main causes leading to hepatic injury, and it is also closely associated with type 2 diabetes, metabolic syndrome, hypertension, and cardiovascular disease. For this reason, a new term from NAFLD, metabolic dysfunction-associated fatty liver disease (“MAFLD”), is considered to be a more suitable overarching term [88]. NAFLD consists of two different conditions: non-alcoholic fatty liver, which is primarily an accumulation of fat in the liver, and non-alcoholic steatohepatitis (NASH), in which fat accumulation is accompanied by inflammation and thus poses a risk of cirrhosis and liver malignancies. Among the various risk factors contributing to NAFLD, changes in gut microbial composition have been recognized as risk factors for NAFLD, obesity, and diabetes [70,89,90]. The consumption of a high-fructose diet promotes increased plasma triglyceride and insulin resistance, leading to increased lipid accumulation in the liver [91]. Previous studies found evidence of a relationship between gut microbiota and bacterial endotoxins in the mechanisms of hepatic steatosis and its progression to NASH [92,93]. SIBO and microbial dysbiosis (increased amounts of *Bacteroides* and *Ruminococcus*, but decreased amount of *Prevotella*) were observed at a higher frequency in patients with NAFLD. Higher amounts of *E. coli* and *B. vulgatus* are associated with fibrosis in stages three and four. The gut microbial signature of patients with NAFLD can be found in Table 3.

In patients with NAFLD and obesity, increased gut barrier permeability develops; thus, metabolic endotoxemia and an increase in the blood levels of LPS occurs. Microbiota-derived LPS has been reported to be correlated with the degree of liver injury and progression to NASH [99].

Several studies have noted the possibility of applying some metabolites to the early diagnosis of NASH. A previous study reported that patients with NASH have increased ethanol produced by their gut microbiota [100]. The gut microbiota can ferment dietary carbohydrates into ethanol, which then enters the blood circulation and is eventually eliminated through the liver. Ethanol-producing bacteria largely include *Bacteroides fragilis*, *Escherichia*, *Bifidobacterium adolescentis*, and *Clostridium thermocellum* [101]. In the gut, ethanol and its metabolites, especially acetaldehyde, lead to the disturbance of tight junctions and increased intestinal permeability, thereby causing gut barrier dysfunction. In the liver, ethanol can cause lipid deposition and inhibit fatty acid ß-oxidation by regulating sterol regulatory element-binding proteins-1c (SREBP-1c) and peroxisome proliferator-activated receptor-α (PPARα) [102]. Ethanol can also aggravate hepatic inflammation and fibrosis by increasing the activity of cytochrome P450 family 2 subfamily E polypeptide 1 (CYP2E1) [103].

In NAFLD, a decrease in bacteria that convert primary bile acids into secondary bile acids is observed. For this reason, the stimulation of bile acid receptors by secondary bile acids is decreased, and further disturbance of the gut microbiota occurs [104]. The composition of the bile acid pool is essential to sustain the diversity of the commensal bacterial community. The detergent effect of bile acids can suppress certain types of bacteria but not others, so the balanced growth of diverse commensal bacteria is properly preserved. FXR and Takeda G protein-receptor-5 (TGR5) are the principal receptors activated by bile acids, and both receptors are under-stimulated because of the decreased DCA [104]. TGR5, a plasma membrane-associated protein, is expressed by cholangiocytes, immune cells, and hepatic stellate cells, and is activated by hydrochloric bile acids. The decreased activation of each receptor leads to hepatic steatosis and chronic inflammatory status.

The gut microbiota can convert choline into trimethylamine (TMA), which is oxidized into TMAO in the host liver. Choline is a water-soluble nutrient essential for biological activities, including maintaining the structural integrity of cell membranes, supporting cholinergic neurotransmission, and donating methyl groups in a number of biosynthetic reactions [105]. TMAO exacerbates impaired glucose tolerance, obstructs hepatic insulin signaling, and promotes adipose tissue inflammation in mice maintained on a high-fat, high-sugar diet [106]. In NAFLD, the conversion of choline into TMA/TMAO is increased; thus, choline deficiency and TMA/TMAO accumulation occur [107,108]. Choline can induce very low-density lipoprotein transportation out of the liver; therefore, choline deficiency leads to the accumulation of lipids in the liver.

Schwiertz et al. reported that the feces of obese individuals have higher levels of SCFAs than that of lean individuals, and this finding suggests that gut microbiota from obese individuals have an enhanced capability to extract and store energy from food compared with lean individuals [109]. The overproduction of SCFAs might promote lipogenesis in the liver because they act as substrates for lipogenesis [110].

### 5.2. Microbiota-Changing Interventions in Non-Alcoholic Fatty Liver Disease

In obese children, supplementation with probiotics improved liver enzymes (alanine aminotransferase, aspartate aminotransferase), triglycerides, and LDL-cholesterol [111]. As shown in Table 4, there have been encouraging results indicating the benefits of probiotics in NAFLD patients [112,113,114]. However, only a few strains/bacterial cocktails have been found to be effective and to slightly improve some of parameters related to NAFLD. Therefore, as shown in a meta-analysis, still more studies are still needed to prove the therapeutic benefit of probiotics in patients with NAFLD [115,116].

Prebiotic treatments, such as fermentable dietary fructo-oligosaccharides, favor the growth of beneficial bacterial species (*Bifidobacterium* spp.), reduce hepatic triglyceride accumulation through the PPARα stimulation of fatty acid oxidation, and lessen cholesterol accumulation by inhibiting SREBP-2-dependent cholesterol synthesis [121]. In addition, prebiotics also increase endogenous intestinotrophic proglucagon-derived peptide production, lower intestinal permeability, and augment tight-junction integrity [122]. Recently, a new study reported that inulin, a kind of indigestible dietary fiber found in an herb, *Jerusalem artichoke*, prevented NAFLD by modulating gut microbiota and suppressing inflammatory pathways, such as LPS-induced TLR4 activation in mice [117].

FMT intervention in high-fat-diet-induced steatohepatitis in mice showed a significant decrease in intrahepatic lipid accumulation and intrahepatic pro-inflammatory cytokines [123]. These results of probiotics, prebiotics, and FMT in NAFLD suggest a positive effect; however, these studies have some limitations due to a lack of high-quality randomized trials.

In addition, obeticholic acid (OCA), the FXR agonist, has been shown to ameliorate NASH in animal models and in patients [124,125]. OCA restored a damaged gut vascular barrier and reduced alanine aminotransferase levels and lipid accumulation in the liver of mice with NASH [126].

## 6. Gut Microbiota in Liver Cirrhosis

### 6.1. Dysbiosis and Microbe-Derived Metabolites in Liver Cirrhosis

Cirrhosis is the end stage of chronic liver diseases, and gut microbiota have also been reported to be altered in patients with liver cirrhosis compared with healthy controls [127,128]. In other words, a deficiency of autochthonous nonpathogenic bacteria and an excessive growth of potentially pathogenic bacteria are commonly observed in patients with liver cirrhosis [127,128,129]. Autochthonous gut taxa include *Lachnospiraceae*, *Ruminococcaceae*, *Veillonellaceae*, and *Clostridiales incertae sedis* XIV, while pathogenic gut taxa include *Staphylococcaceae*, *Enterobacteriaceae*, and *Enterococcaceae* [129]. The gut microbial signature of patients with alcoholic liver cirrhosis or NAFLD cirrhosis can be found in Table 5. The gut microbiota profile of patients with alcoholic liver cirrhosis had an increased relative abundance of *Enterobacteriaceae* and a decreased relative abundance of *Lachnospiraceae* and *Ruminococcaceae*. Additionally, a greater increase in the levels of oral-origin microbiota in the stool was reported in patients with alcoholic cirrhosis than in those without cirrhosis [130]. The increase in oral microbiota in the stool in those with cirrhosis, especially in patients with alcoholic cirrhosis, is probably an epiphenomenon given the high rate of periodontitis, the change in salivary microbiota, proton pump inhibitor use, and relatively low gastric acid levels in these patients [128,131,132,133]. NAFLD cirrhosis was associated with increases in the levels of *Veillonella parvula*, *Veillonella atypica*, *Ruminococcus gnavus*, *Clostridium bolteae*, and *Acidaminococcus* spp., and accompanied by decreases in the abundances of *Eubacterium eligens*, *Eubacterium rectale*, and *Faecalibacterium prausnitzii*. *Veillonella parvula* and *Faecalibacterium prausnitzii* were reported to be the most critical species for discriminating between NAFLD cirrhosis and the control group [134].

Changes in the microbiota composition in liver cirrhosis are derived from reduced small bowel motility, bile acid abnormalities, and impaired intestinal immunity. Ascites is one of the key contributors to delaying gut transit and developing dysbiosis in cirrhotic patients [136,137]. An altered bile acid pool exhibits reduced primary bile acid levels and increased secondary bile acid levels in the gut [135,138,139]. SIBO is commonly observed in cirrhotic patients as a result of decreased intestinal motility and delayed transit times, and the deterioration of liver cirrhosis is related to SIBO [140]. The severity of SIBO can be connected to the degree of deterioration of the liver cirrhosis status [141]. Increased intestinal permeability may promote bacterial translocation into systemic circulation. SIBO is known as a main risk factor in the etiology of both spontaneous bacterial peritonitis (SBP) and hepatic encephalopathy (HE) in cirrhotic patients [142,143]. In cirrhotic patients, HE and SBP commonly occur. Ammonia and endotoxins that are produced by urease-producing bacteria, such as *Klebsiella* and *Proteus*, are known as the main contributors to the development of HE [144]. Disturbed intestinal barriers and increased microbial translocation are involved in the mechanism of SBP development [145]. LPS is a main product from *Escherichia*/*Shigella*, and the overgrowth of these bacteria can lead to increased intestinal permeability and cause endotoxemia, which is related to deteriorating disease severity and complications in cirrhosis [146].

One of the proposed mechanisms of dysbiosis in patients with cirrhosis pertains to the decreased production of bile acids. *Ruminococcaceae* is one of few taxa known to contain secondary bile acid-producing bacteria, and it has been reported that there is a positive correlation between the abundance of *Ruminococcaceae* and DCA production [135]. Most bile acids are reabsorbed in the terminal ileum and transported back to the liver through gut–liver circulation. However, some bile acids reach the colon and are converted by the gut microbiota into secondary bile acids. Secondary bile acids modulate functions connected with glucose and fat metabolism in the liver [147,148]. *Lactobacillus*, *Bifidobacteria*, *Enterobacter*, *Bacteroides*, and *Clostridium* are related to secondary bile acid formation. A decreased conversion of primary to secondary bile acids is caused by dysbiosis in cirrhotic patients [135,149]. Reduced bile flow, decreased fecal bile acids, and increased serum bile acids are characteristics of cirrhosis that also deteriorate in proportion with cirrhosis severity [135,138,139]. Liver dysfunction can lead to the impairment of the synthesis and excretion of bile acids; as a result, decreased levels of total bile acids in the gut and increased levels in the serum are observed. Additionally, decreased bile flow leads to diminished intestinal FXR signaling, which disturbs intestinal barrier function by decreasing mucous thickness and antimicrobial peptide synthesis, and injuring the gut vascular barrier.

There are some similarities and some unique differences in the gut microbiota composition and metabolites in patients with ALD, NAFLD, and cirrhosis (Figure 1).

### 6.2. Microbiota-Changing Interventions in Liver Cirrhosis

Recently, in an international cirrhosis cohort study, Bajaj et al. reported that a diet rich in fermented milk, vegetables, cereals, coffee, and tea was associated with greater microbial diversity, and this was linked to a lower risk of long-term hospitalization [150]. Dairy proteins and vegetable proteins contribute to a decrease in serum ammonia levels due to increasing ammonia detoxification through the urea cycle and accelerating intestinal transit by their high fiber content. Vegetable proteins also lead to reduced circulating mercaptans and indoles [151,152,153]. Diet control may be one of the treatment strategies for gut microbiota modification, but the results are not promising due to poor compliance.

Probiotics that provide beneficial effects to liver cirrhosis can be found in Table 6. Shi et al. reported that *L. salivarius* or *P. pentosaceus* extended the survival time and considerably ameliorated carbon tetrachloride (CCl4)-induced liver cirrhosis in rats [154]. They found that *L. salivarius* or *P. pentosaceus* achieved this effect through the alleviation of gut microbiota dysbiosis, an improvement in intestinal barrier function, decreased bacterial transformation, and reduced liver inflammatory response. In another study using CCl4-induced experimental cirrhosis, a probiotic mixture (VSL#3: *S. thermophiles*, *B. breve*, *B. longum*, *B. infantis*, *L. paracasei*, *L. acidophilus*, *L. delbrueckii* ssp. *bulgaricus* and *L. plantarum*) decreased bacterial translocation, the proinflammatory state (decrease in TNF-α levels), and ileal oxidative damage, and increased ileal expression of the tight junction protein occludin [155]. In addition, *A. muciniphila* showed beneficial effects on immune-mediated liver injury in C57BL/6 mice by alleviating inflammation and hepatocellular death [156].

Bajaj et al. reported that oral FMT capsules were safe and well tolerated in in patients with cirrhosis and recurrent HE, and that they improved duodenal mucosal diversity, dysbiosis, and cognitive function [160]. In addition, they found that oral capsular FMT led to microbial functional changes (higher secondary/primary fecal and serum bile acid ratios) and the improvement of cognitive function in randomized, placebo-controlled trial [161]. However, larger studies are needed to confirm the beneficial FMT effect on microbial functional change and longer-term outcomes after FMT in cirrhosis.

## 7. Conclusions

Chronic liver disease with diverse etiologies progresses to liver fibrosis through multiple common mechanisms of pathogenesis. The altered gut microbiome is one of the main mechanisms that are shared within various liver disease etiologies. Several common and overlapping important pathophysiologic processes include: (1) SIBO and gut dysbiosis; (2) gut barrier dysfunction and abnormal intestinal permeability; (3) changes in primary and secondary bile acid profiles; and (4) alteration of microbial-produced metabolites caused by gut dysbiosis (Figure 2).

An understanding of the gut–liver axis has advanced in the last decade. It has been confirmed that there is a strong connection between the gut microbiota and the liver, the so-called “gut–liver axis”, which acts as an important contributor to the pathogenesis of chronic liver diseases. The coexisting microorganisms, with the majority living in the digestive tract from where they produce or modify various chemicals, or trigger host reactions that affect various physiological functions and pathologies. However, the precise mechanism of this connection in diverse liver diseases is still uncertain. Most studies are snapshots of microbiome landscapes; therefore, more expanded knowledge is needed on the short-term and long-term dynamics of the intestinal microbiome.

In the studies using probiotics, prebiotics, and FMT, the modulation of the intestinal microbiota can potentially be a preventive and therapeutic approach for chronic liver disease. To identify candidate microorganisms for therapy, appropriate preclinical models are necessary. Although human and murine gut floras share 90% and 89% similarities in phyla and genera, respectively [162], there are key discrepancies in the makeup and abundance of microbes, representatively the *Firmicutes*/*Bacteroidetes* (F/B) ratio (with humans having a greater F/B ratio, whereas the inverse is true for mice) [163,164,165]. Therefore, the establishment of a humanized gnotobiotic mouse model through the FMT of human feces into germ-free mice provides a powerful tool to mimic the human microbial system. Upon transplantation into germ-free mice, microbial species of the human microbiota are affected by a diet dissimilar to the human donor. However, by employing diet questionnaires to fecal donors and customizing corresponding research diets, there is an opportunity to take humanized murine microbiota studies to higher precision and translatability.

Finally, to standardize microbiota modulation therapy and guide personal everyday health behavior or clinical practice, large prospective randomized controlled studies of long duration are required. Although both the gut microbiota composition and gut microbe-produced metabolites have been tested as therapeutic targets, we have a long way to go before achieving mechanistic insights into how the gut microbiome mediates or modifies various liver diseases. Future studies on the interactions within the global intestinal microbial community, including fungi, bacteriophages, and eukaryotic virus, are needed. In addition, the integration of microbiome data with other omics data and bio-clinical variables is required. These efforts will give hope for a new therapeutic strategy against chronic liver disease.

## Figures and Tables

**Figure 1 ijms-23-00426-f001:**
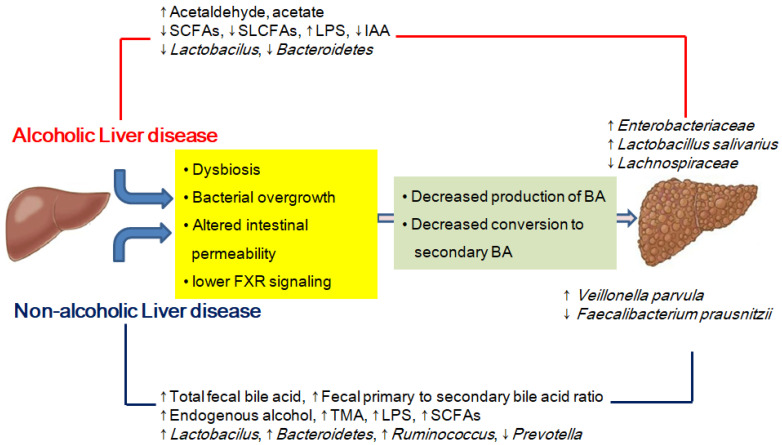
Similarities and some unique differences in the gut microbiota composition and metabolites in patients with ALD, NAFLD, and cirrhosis. SCFAs, short-chain fatty acids; SLCFAs, saturated long-chain fatty acids; LPS, lipopolysaccharide; IAA, indole-3-acetic acid; FXR, farnesoid X receptor; BA, bile acid; TMA, trimethylamine.

**Figure 2 ijms-23-00426-f002:**
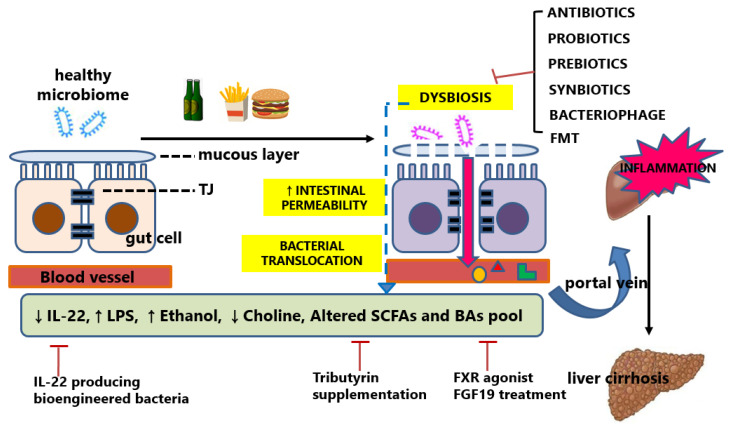
Dysbiosis and therapeutic intervention in chronic liver disease. The defective intestinal barrier due to the malfunction of the tight junctions promotes the translocation of bacterial products into the portal vein. Therapeutic interventions targeting gut microbiota composition or their metabolites have been attempted. TJ, tight junction; IL-22, interleukin-22; LPS, lipopolysaccharide; SCFAs, short-chain fatty acids; BAs, bile acids; FXR, farnesoid X receptor; FGF19, fibroblast growth factor 19.

**Table 1 ijms-23-00426-t001:** Dysbiosis associated with alcoholic liver disease.

Conditions		Methods	Main Results (Phylum_Taxon)	Ref.
Human	LC with (*n* = 17) or without sAH (*n* = 17)	16S ribosomal RNA sequencing	LC with sAH: ↑ *Actinobacteria*, ↓ *Bacteroidetes*	[35]
Human	HC (*n* = 24) HDC (*n* = 20) mAH (*n* = 10) sAH (*n* = 24)	16S ribosomal RNA sequencing	HDC vs. HC: ↓ *Bacteroidetes*, ↑ *Firmicutes* to *Bacteroidetes* ratio ↑ *Proteobacteria*_*Enterobacteriaceae*, ↑ *Firmicutes*_*Lachnospiraceae*, ↑ *Firmicutes*_*Lactobacillaceae*, ↑ *Firmicutes*_*Streptococcaceae*, ↑ *Bacteroidetes*_*Prevotellaceae*, ↑ *Saccharibacteria*, ↑ *Firmicutes*_*Veillonellaceae* sAH vs. HC: ↑ *Proteobacteria* AH vs. HDC: ↑ *Firmicutes*_*Veillonella*, ↑ *Bacteroidetes*_*Bacteroides*	[36]
Human	AH (*n* = 74)	16S ribosomal RNA sequencing	AH: ↓ *Verrucomicrobia*_*Akkermansia*, ↑ *Firmicutes*_*Veillonella* ↓ *Bacteroidetes*_*Bacteroides*	[37]
Human	AUD (*n* = 36) LC (*n* = 14)	16S ribosomal RNA sequencing	AUD vs. Control: ↓ *Verrucomicrobia*_*Akkermansia*, ↑ *Bacteroidetes*_*Bacteroides*, ↑ serum LPS,	[38]
			↑ TNF—α, IL1β, monocyte chemoattractant protein LC vs. non-LC: ↑ IL6, ↑ IL8	

LC, liver cirrhosis; sAH, severe alcoholic hepatitis; HC, healthy control; HDC, heavy drinking control; mAH, moderate alcoholic hepatitis; AH, alcoholic hepatitis; AUD, alcohol use disorder; LPS, lipopolysaccharide; TNF, tumor necrosis factor; IL, interleukin.

**Table 2 ijms-23-00426-t002:** Treatment targeting dysbiosis in alcoholic liver disease.

Conditions		Treatment	Main Results	Ref.
C57BL/6J mice	Chronic feeding: 25 days, 32% of total kcal Short-term ethanol feeding: 2 days, 32% of total kcal Acute single gavage (5 g/kg)	Tributyrin (butyrate supplementation), 0.83–10 mM (liquid diet or oral gavage)	(1) Protective effect to tight junction proteins, butyrate receptor and transporter (2) Mitigation of inflammatory measures	[78]
C57BL/6J mice	EtOH group: 5% *v/v* ethanol-containing diet for 10 days + single ethanol gavage (5 g/kg) Control mice: isocalorically pair-fed maltose dextrin	Tributyrin: 5 mM	Mitigation of ethanol effect-↓ disruption of intestinal tight junction localization and intestinal permeability, liver injury	[79]
C57BL/6J mice (*Atp4a ^Sl/Sl^* mice)	EtOH group: Lieber-DeCarli diet containing 36% ethanol for 10 days	Bacteriophages targeting cytolytic *Enterococcus faecalis*	Decrease cytolysin in the liver Abolish ethanol-induced liver disease and steatosis Reduced fecal amounts of *Enterococcus*	[80]
C57BL/6J mice	EtOH group: Lieber-DeCarli alcohol for 8 weeks Control group: isocaloric control diet	Fexaramine (intestine-restricted FXR agonist): 100 mg/kg daily during 8 weeks of alcohol	Fexaramine treatment group -stabilize the gut barrier - modulate hepatic genes involved in lipid metabolism	[81]
		or AVV expressing the human nontumorigenic FGF19-variant M52	FGF19 treatment group - ameliorate alcoholic steatohepatitis	
C57BL/6J mice	Chronic-binge ethanol diet Control group: isocaloric control diet	IAA (gavage of 100 μL of 20 mM IAA), or	IAA treatment group -↓ liver damage and steatosis -↓ ALT and hepatic levels of triglyceride -↓ ethanol-induced bacterial translocation	[82]
		Engineered bacteria: *Lactobacillus reuteri*/IL-22	Engineered bacteria treatment group -restore intestinal levels of IL-22 -re-expression of REG3G -↓ bacterial translocation and ethanol-induced steatohepatitis	
C57BL/6J mice	EtOH group: Lieber-DeCarli diet containing 5–6% ethanol for 10 or 14 days	Synthetic TLR7 ligand 1Z1 Orally administrated 1µmol or subcutaneous injection 0.4 µmol	↓ Intestinal barrier disruption and bacterial translocation ↑ Expression of antimicrobial peptides, REG3B and REG3G	[83]
			Modulate the microbiome	

EtOH, ethanol; FXR, farnesoid X receptor; AVV, adeno-associated virus; FGF, fibroblast growth factor; IAA, indole-3-acetic acid; ALT, alanine aminotransferase; IL, interleukin; REG3G, regenerating islet-derived 3 gamma; TLR, Toll-like receptor; IL, interleukin.

**Table 3 ijms-23-00426-t003:** Dysbiosis associated with non-alcoholic fatty liver disease.

Conditions		Methods	Main Results (Phylum_Taxon)	Ref.
Human	NAFLD (*n* = 57) : NASH (*n* = 35) vs. No NASH (*n* = 22) or F0/F1 (*n* = 30) vs. F2 ≤ fibrosis (*n* = 27)	16S ribosomal RNA sequencing	NASH and F2 ≤ fibrosis: ↑ *Bacteroidetes*_*Bacteroides*, ↓ *Bacteroidetes*_*Prevotella*	[94]
			F2 ≤ fibrosis: ↑ *Firmicutes*_*Ruminococcus*	
Human	NAFLD (*n* = 25) vs. Healthy (*n* = 22) NASH vs. No NASH F0/1 vs. F2 ≤ fibrosis	16S rDNA amplicon sequencing	NAFLD: ↑ *Proteobacteria*, ↑ *Fusobacteia*, ↓ *Bacteroidetes*	[95]
			NASH: ↑ *Firmicutes*_*Blautia*, ↑ *Firmicutes*_*Lachnospiraceae* F2 ≤ fibrosis: ↑ *Proteobacteria_Escherichia*, ↑ *Proteobacteria*_*Shigella*	
Human	NAFLD (*n* = 13) : Biopsy-proven NASH (lean 4, overweight 5, obese 4) Control (*n* = 10)	16S ribosomal RNA sequencing	F ≤ fibrosis: ↑ *Firmicutes*_*Lactobacilli*	[96]
			Lean NASH: ↓ *Firmicutes*_*Faecalibacterium*, ↓ *Firmicutes*_*Ruminococcus* Obese NASH: ↑ *Firmicutes*_*Lactobacilli* Overweight NASH: ↓ *Actinobacteria Bifidobacterium*	
Human	Obese NAFLD (*n* = 36) Obese patients without NAFLD (*n* = 17) Healthy control (*n* = 20)	16S ribosomal RNA sequencing	NAFLD: ↑*Fermicutes*, ↑ *Fermicutes_Streptococcus*	[97]
			Obese with or without NAFLD : ↓ *Firmicutes*_*Blautia*, *Firmicutes*_*Alkaliphilus*, *Bacteroidetes*_*Flavobacterium*, *Verrucomicrobia*_*Akkermansia*	
Human	NAFLD (*n* = 25) NASH (*n* = 25) Healthy control (*n* = 25)	16S ribosomal RNA sequencing	NAFLD: ↑*Bacteroidetes*, ↓ *Firmicutes*	[98]
			NAFL or NASH: ↓ *Firmicutes*_*Ruminococcaceae* UCG-010, *Firmicutes*_*Ruminococcaceae*, *Firmicutes*_*Clostridiales*, and *Firmicutes*_*Clostridia*	

NAFLD, non-alcoholic fatty liver disease; NASH, non-alcoholic steatohepatitis; NAFL, non-alcoholic fatty liver.

**Table 4 ijms-23-00426-t004:** Treatment targeting dysbiosis in non-alcoholic fatty liver disease.

Conditions		Treatment	Main Results	Ref.
Human	Randomized, triple blind trial (64 children with NAFLD)	Probiotic capsule (*L. acidophilus*, *B. bifidum*, *B. lactis*, *L. rhamnosus*), 12 weeks or placebo	Probiotic treatment group - enhanced liver enzyme -↓ mean cholesterol, LDL-cholesterol, triglyceride, waist circumference	[111]
Human	Open-label, randomized controlled clinical trial (*n* = 102)	300 g synbiotic yogurt (*B. animalis*, inulin)	Synbiotic yogurt consumption group - improved hepatic steatosis and liver enzyme	[112]
		or conventional yogurt, 24 weeks or control group		
Human	Obese NAFLD (*n* = 68)	Probiotic mixture (*L. acidophilus*, *L. rhamnosus*, *Lacticaseibacillus paracasei*, *P. pentosaceus*, *B. lactis*, and *B. breve*) 12 weeks or placebo	Probiotic treatment -↓ body weight and total body fat	[113]
Human	NAFLD (*n* = 89)	Probiotic (*L. casei*, *L. rhamnosus*, *L. acidophilus*, *B. longum*, and *B. breve*) or prebiotic (oligofructose), 12 weeks or placebo	Probiotics group -↓ triglyceride, AST, ALT, GGT, ALP	[114]
			Prebiotics group -↓ triglyceride, LDL-cholesterol, AST, ALT	
C57BL/6J mice		ND/HFD/ND + inulin/HFD + inulin for 14 weeks	Inulin treatment group -restored abnormal indicators observed in HFD group -reduced TLR4 + hepatic macrophages, NF-κB, nod-like receptor protein 3, apoptosis-associated speck-like protein and caspase-1 -↑ *Akkermansia, Bifidobacterium* ↓ *Blautia,* the ratio of *Firmicutes*/*Bacteroidetes* - increase short-chain fatty acids	[117]
Human	14 patients with liver biopsy-confirmed NASH	Randomized to receive oligofructose (8 g/day) for 12 weeks followed by 16 g/day for 24 weeks or isocaloric placebo for 9 months	Oligofructose improved liver steatosis and overall NAS score	[118]
Human	Adult with definite NASH, NAS score ≥ 4, F2-3 or F1 with at least one accompanying comorbidity	Randomly assigned in 1:1:1 oral placebo: *n* = 311 obeticholic acid 10 mg: *n* = 312 obeticholic acid 25 mg: *n* = 308	Obeticholic acid at 25 mg significantly improved fibrosis and key components of NASH disease activity	[119]
Human	NAFLD patients, *n* = 198	Randomly assigned in 1:1:1 placebo (*n* = 64) norursodeoxycholic acid 500 mg/day (*n* = 67) norursodeoxycholic acid 1500 mg/day (*n* = 67) for 12 weeks	Norursodeoxycholic acid at 1500 mg resulted in a significant reduction of serum ALT within 12 weeks	[120]

NAFLD, non-alcoholic fatty liver disease; LDL, low-density lipoprotein; AST, aspartate aminotransferase; ALT, alanine aminotransferase; GGT, gamma-glutamyl transferase; ALP, alkaline phosphatase; TLR, Toll-like receptor; NF-Κb, nuclear factor kappa-light-chain-enhancer of activated B cells; ND, normal diet; HFD, high fat diet; NASH, non-alcoholic steatohepatitis; NAS, non-alcoholic fatty liver activity score.

**Table 5 ijms-23-00426-t005:** Dysbiosis associated with liver cirrhosis.

Conditions	Methods	Main Results (Phylum_Taxon)	Ref.
Alcoholic LC (*n* = 43) vs. other etiologies (*n* = 170)	Multi-tagged pyrosequencing	Alcoholic LC: ↑ *Proteobacteria*_*Enterobacteriaceae*, ↑ *Proteobacteria*_*Halomonadaceae*, ↓ *Firmicutes*_*Lachnospiraceae*, ↓ *Firmicutes*_*Ruminococcaceae*, ↓ *Firmicutes*_*Clostridialies* XIV	[129]
Alcohol dependence with LC vs. Alcohol dependence (*n* = 27) without LC (*n* = 72)	Shotgun metagenomic analysis	Alcoholic LC: ↑ *Firmicutes*_*Lactobacillus salivarius*, ↑ *Firmicutes*_*Veillonella parvula*, ↑ *Firmicutes*_*Streptococcus salivarius*, ↑ *Actinobacteria*_*Bifidobacterium*	[130]
NASH cirrhosis (*n* = 32) vs. other etiologies (*n* = 181)	Multi-tagged pyrosequencing	NASH cirrhosis: ↑ *Bacteroidetes*_*Porphyromonadaceae*, ↑ *Bacteroidetes*_*Bacteroidaceae*, ↓ *Firmicutes*_*Veillonellaceae*	[129]
Non-NAFLD group (*n* = 54) vs. NAFLD-cirrhotic group (*n* = 27)	Shotgun metagenomic analysis	NAFLD cirrhotic group: ↑ *Firmicutes*_*Veillonella parvula*, ↑ *Firmicutes*_*Veillonella atypic**a*, ↑ *Firmicutes*_*Ruminococcus gnavus*, ↑ *Firmicutes*_*Clostridium bolteae*, ↑ *Firmicutes*_*Acidaminococcus* spp., ↓ *Firmicutes*_*Eubacterium eligens*, ↓ *Firmicutes*_*Eubacterium rectal*, ↓ *Firmicutes*_*Faecalibacterium prausnitzii*	[134]
LC of multiple aetiology (*n* = 36) vs. Healthy control (*n* = 24)	16S ribosomal RNA sequencing	LC: ↓ *Bacteroidetes*_*Bacteroidetes*, ↓ *Proteobacteria*, ↓ *Fusobacteria*, ↑ *Proteobacteria_ Enterobacteriaceae*, ↑ *Firmicutes*_*Veillonellaceae*, ↑ *Firmicutes*_*Streptococcaceae*, ↓ *Firmicutes*_*Lachnospiraceae*	[43]
LC of multiple aetiology (*n* = 47) vs. Control (*n* = 14)	16S ribosomal RNA sequencing	LC: ↑ *Proteobacteria_Enterobacteriaceae*, ↓ *Firmicutes*_*Lachnospiraceae*, ↓ *Firmicutes*_*Ruminococcaceae*, ↓ *Firmicutes*_*Blautia*	[135]

LC, liver cirrhosis; NAFLD, non-alcoholic fatty liver disease; NASH, non-alcoholic steatohepatitis.

**Table 6 ijms-23-00426-t006:** Treatments targeting dysbiosis in liver cirrhosis.

Conditions		Treatment	Main Results	Ref.
Human	Double blind trial, LC with an episode of HC during the previous month	Probiotic preparation (VSL #3) (*n* = 66) 6 months or placebo (*n* = 64)	↓ The risk of hospitalization for HE ↓ CTP and MELD score	[157]
Human	LC with MHE : randomized LGG or placebo group	Probiotic group, 8 weeks or placebo	↓ Endotoxemia, TNF-α	[158]
			Alteration in gut microbiome -↓ *Enterobacteriaceae,* ↑ *Clostridiales Incertae Sedis* XIV, ↑ *Lachnospiraceae*	
Rat	CCl4-induced cirrhotic rats	LI01: *L. salivarius* LI05: *P. pentosaceus* *L. rhamnosus* GG, *C. butyricum* MIYAIRI and *Bacillus*, 13 weeks	LI01 or LI05	[154]
			-prevent liver fibrosis, ↓ hepatic expression of profibrogenic genes	
			Alteration in gut microbiome	
			-↓ *Enterobacteriaceae*, ↑ *Clostridiales Incertae Sedis* XIV and *Lachnospiraceae*	
C57BL/6J mice	Bile-duct ligation CCl4-induced cirrhosis	FXR-agonists oral gavage : fexaramine (100 mg/kg/day) obeticholic acid (30 mg/kg/day)	FXR-agonists treatment group - ameliorate pathological translocation of GFP-*E. coli* from the ileal lumen to the liver in cirrhotic mice Obeticholic acid treatment group -significantly increases ileal TJ protein expression (ZO1, claudin-1,-2, and occludin), upregulating them to the level of healthy control mice	[159]

LC, liver cirrhosis; HC, hepatic encephalopathy; CTP, Child–Turcotte–Pugh; MELD, model for end-stage liver disease; MHE, minimal hepatic encephalopathy; CCl4, carbon tetrachloride; TNF, tumor necrosis factor; FXR, farnesoid X receptor; TJ, tight junction; GFP-*E. coli*, green fluorescent protein, *Escherichia coli*.

## Data Availability

Not applicable.

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
