# Peer review of "Role of Microbiota-Derived Metabolites in Alcoholic and Non-Alcoholic Fatty Liver Diseases"

_ijms, 2021, doi:10.3390/ijms23010426_

Round 1
Reviewer 1 Report
In this review, the authors have discussed the metabolites generated by gut microbiota that are transported to the liver through venous branches of the portal vein and contribute to the pathology of chronic liver diseases, including alcoholic liver disease and non-alcoholic fatty liver disease, and liver cirrhosis. The authors have discussed in detail the types of metabolites, their sources, and associated microbes for all three chronic liver diseases. Moreover, the authors have also talked about the interventions targeting the gut microbiome to treat chronic liver diseases. The review is well written and I recommend the publication of this review.
Author Response
I sincerely appreciate your favorable review.
Reviewer 2 Report
In the presented review, Park and colleagues provide a comprehensive overview of microbiome alterations in alcoholic liver disease (ALD), non-alcoholic fatty liver disease (NAFLD) and in cirrhosis. Overall, this summary is timely and informative. However, multiple aspects of the work appear somewhat superficial and should, in my opinion, be revised prior to publication.
Specific comments:
- The inclusion of cirrhosis as a separate entity besides ALD and NAFLD appears somewhat inaccurate. I would assume that the observed microbiomal changes should differ considerably depending on the etiology of cirrhosis. For instance, it would be unexpected if cirrhosis downstream of NAFLD would result in similar dysbiosis as cirrhosis due to HCV infection. However, Table 5 does not distinguish by etiology. Furthermore, it appears that the authors did not even consider these differences in their discussion.
- Abstract: “release a wide number of metabolites including pathogen-associated molecular pattern molecules, peptides, and hormones” – Why do the authors omit short-chain fatty acids here, which they later list as the first key metabolite of the microbiota (p. 3). Also, PAMPs are conventionally not referred to as “metabolites”.
- ”In parts, the review is written for a very basic audience. For instance statements such as “the assemblage of living microorganisms in a defined environment is called the microbiota” should in my opinion be removed as too basic.
- “The gut microbiota has a cell number similar with human, however genes of microbiota are 450-fold more than that of human” – What should this mean?
- Tables 1, 3 and 5 are informative. However, it would help the reader if the changes in microbiome could be provided at both the level of phyla, classes and orders. As is, the liver-inclined readership who might not be fully familiar with bacterial phylogenetic relationships will have a hard time in recognizing patterns. For instance, do Akkermansia and Veillonella belong to the Bacteriodetes or to the Firmicutes, and are the different studies thus in agreement or do they report divergent results?
- The translatability of microbiota studies should be critically discussed. To be discussed in this context, e.g.: How similar are the human and the murine microbiome? What are effects of a controlled semi-sterile lab environment on microbial complexity? What are the consequences of coprophagia typically seen in rodents?
- I would like to see a direct comparison of microbiomic alterations in AFLD, NAFLD and cirrhosis in a Figure. What is similar what is different?
- Contrary to the title, which suggests that the manuscript focusses on “microbiota-derived metabolites”, all tables focus on the bacterial composition itself without discussing highlighting in SCFA levels, PAMPs etc. It would be very helpful if the authors would add this information, e.g. by adding an additional column “measured changes in microbial metabolites” to the tables.
Author Response
Specific Comments:
- Comment 1: The inclusion of cirrhosis as a separate entity besides ALD and NAFLD appears somewhat inaccurate. I would assume that the observed microbiomal changes should differ considerably depending on the etiology of cirrhosis. For instance, it would be unexpected if cirrhosis downstream of NAFLD would result in similar dysbiosis as cirrhosis due to HCV infection. However, Table 5 does not distinguish by etiology. Furthermore, it appears that the authors did not even consider these differences in their discussion.
- Response 1: Thanks for your valuable comments. As your opinion, we have added content about alcoholic liver cirrhosis and NAFLD-cirrhosis to the ‘6.1. Dysbiosis and microbe-derived metabolites in liver cirrhosis’ section (page 11). Also, we changed the Table 5 (page 11-12).
- Comment 2: Abstract: “release a wide number of metabolites including pathogen-associated molecular pattern molecules, peptides, and hormones” – Why do the authors omit short-chain fatty acids here, which they later list as the first key metabolite of the microbiota (p. 3). Also, PAMPs are conventionally not referred to as “metabolites”.
- Response 2: Thanks for your valuable comments. Based on what you pointed out, we revised the abstract as follows.
“The gut microbiota consists of various microorganisms that have a role of maintaining the homeostasis of the host and release a wide number of metabolites including short-chain fatty acids (SCFAs), peptides, and hormones, continually shaping host immunity and metabolism. The integrity of intestinal mucosal and vascular barriers is crucial to protect liver cells from exposure to harmful metabolites and pathogen-associated molecular pattern molecules.” (page 1)
- Comment 3: ”In parts, the review is written for a very basic audience. For instance statements such as “the assemblage of living microorganisms in a defined environment is called the microbiota” should in my opinion be removed as too basic.
- Response 3: Thanks for your comment. As your suggestion, we removed the sentence “the assemblage of living microorganisms in a defined environment is called the microbiota” (page 2).
- Comment 4: “The gut microbiota has a cell number similar with human, however genes of microbiota are 450-fold more than that of human” – What should this mean?
- Response 4: Thanks for your comment. In the previous study, the number of bacteria in the human gut has been estimated to exceed the number of somatic cells in the body and that the biomass of the gut microbiota may reach up to 1.5 kg. Recent findings have demonstrated that the gut microbiome complements our human genome with at least 450-fold more genes. To clarify the meaning, we changed the sentence as follow.
“The gut microbiota has a cell number similar with human. Furthermore, the combined genomes of the gut microbiota-the microbiome-contain 450-fold more genes than are encoded in the human genome.” (page 3)
- Comment 5: Tables 1, 3 and 5 are informative. However, it would help the reader if the changes in microbiome could be provided at both the level of phyla, classes and orders. As is, the liver-inclined readership who might not be fully familiar with bacterial phylogenetic relationships will have a hard time in recognizing patterns. For instance, do Akkermansia and Veillonella belong to the Bacteriodetes or to the Firmicutes, and are the different studies thus in agreement or do they report divergent result?
- Response 5: Thanks for your comment. As your suggestion, we changed Table 1,3 and 5. We described microbiota in the form ( Phylum_Taxon).
- Comment 6: The translatability of microbiota studies should be critically discussed. To be discussed in this context, e.g.: How similar are the human and the murine microbiome? What are effects of a controlled semi-sterile lab environment on microbial complexity? What are the consequences of coporphagia typically seen in rodents?
- Response 6: Thanks for your valuable comment. Although the general anatomy of mice and humans is similar, distinctions remain in regard to the structural design of the gastrointestinal tract As your opinion, we added the content as follow.
“To identify candidate microorganisms for therapy, appropriate preclinical models are necessary. Although human and murine gut floras share 90% and 89% similarities in phyla and genera, respectively [162], there are key discrepancies in the makeup and abundance of microbes, representatively the Fermicutis/Bacteroidetes (F/B) ratio (human having a greater F/B ratio, whereas the inverse is true for mice) [163-165]. Therefore, the establishment of a humanized gnotobiotic mouse model through FMT of human feces into germ-free mice provide a powerful tool to mimic the human microbial system. Upon transplantation into germ-free mice, microbial species of the human microbiota are affected by a diet dissimilar with human donor. However, by employing diet questionnaires to fecal donors and customizing corresponding research diets, there is an opportunity to take humanized murine microbiota studies to higher precision and translatability. (page 15)
- Comment 7: I would like to see a direct comparison of microbiomic alterations in AFLD, NAFLD and cirrhosis in in a Figure. What is similar what is different?
- Response 7: Thanks for your comment. As your comment, we showed some similarity and some unique differences in the gut microbiota composition and metabolites in patients with ALD, NAFLD and cirrhosis in Figure 1. (page 13)
- Comment 8: Contrary to the title, which suggests that the manuscript focusses on “microbiota-derived metabolites”, all tables focus on the bacterial composition itself without discussing highlighting in SCFA levels, PAMPs etc. It would be very helpful if the authors would add this information, , e.g. by adding an additional column “measured changes in microbial metabolites” to the tables.
- Response 8: Thanks for your comment. We were concerned that adding a column will complicate the table too much, so we showed the change in metabolite in the newly created Figure 1. I ask for your understanding. (page 13)